# Machine Learning and Deep Learning Methods for Intrusion Detection Systems: A Survey

**Hongyu Liu * and Bo Lang**

State Key Laboratory of Software Development Environment, Beihang University, Beijing 100191, China; langbo@buaa.edu.cn

**\*** Correspondence: liuhongyu@buaa.edu.cn

**Abstract:** Networks play important roles in modern life, and cyber security has become a vital research area. An intrusion detection system (IDS) which is an important cyber security technique, monitors the state of software and hardware running in the network. Despite decades of development, existing IDSs still face challenges in improving the detection accuracy, reducing the false alarm rate and detecting unknown attacks. To solve the above problems, many researchers have focused on developing IDSs that capitalize on machine learning methods. Machine learning methods can automatically discover the essential differences between normal data and abnormal data with high accuracy. In addition, machine learning methods have strong generalizability, so they are also able to detect unknown attacks. Deep learning is a branch of machine learning, whose performance is remarkable and has become a research hotspot. This survey proposes a taxonomy of IDS that takes data objects as the main dimension to classify and summarize machine learning-based and deep learning-based IDS literature. We believe that this type of taxonomy framework is fit for cyber security researchers. The survey first clarifies the concept and taxonomy of IDSs. Then, the machine learning algorithms frequently used in IDSs, metrics, and benchmark datasets are introduced. Next, combined with the representative literature, we take the proposed taxonomic system as a baseline and explain how to solve key IDS issues with machine learning and deep learning techniques. Finally, challenges and future developments are discussed by reviewing recent representative studies.

**Keywords:** machine learning; deep learning; intrusion detection system; cyber security

---

## 1. Introduction

Networks have increasing influences on modern life, making cyber security an important field of research. Cyber security techniques mainly include anti-virus software, firewalls and intrusion detection systems (IDSs). These techniques protect networks from internal and external attacks. Among them, an IDS is a type of detection system that plays a key role in protecting cyber security by monitoring the states of software and hardware running in a network.

The first intrusion detection system was proposed in 1980 [1]. Since then, many mature IDS products have arisen. However, many IDSs still suffer from a high false alarm rate, generating many alerts for low nonthreatening situations, which raises the burden for security analysts and can cause seriously harmful attack to be ignored. Thus, many researchers have focused on developing IDSs with higher detection rates and reduced false alarm rates. Another problem with existing IDSs is that they lack the ability to detect unknown attacks. Because network environments change quickly, attack variants and novel attacks emerge constantly. Thus, it is necessary to develop IDSs that can detect unknown attacks.

To address the above problems, researchers have begun to focus on constructing IDSs using machine learning methods. Machine learning is a type of artificial intelligence technique that can

automatically discover useful information from massive datasets [2]. Machine learning-based IDSs can achieve satisfactory detection levels when sufficient training data is available, and machine learning models have sufficient generalizability to detect attack variants and novel attacks. In addition, machine learning-based IDSs do not rely heavily on domain knowledge; therefore, they are easy to design and construct. Deep learning is a branch of machine learning that can achieve outstanding performances. Compared with traditional machine learning techniques, deep learning methods are better at dealing with big data. Moreover, deep learning methods can automatically learn feature representations from raw data and then output results; they operate in an end-to-end fashion and are practical. One notable characteristic of deep learning is the deep structure, which contains multiple hidden layers. In contrast, traditional machine learning models, such as the support vector machine (SVM) and k-nearest neighbor (KNN), contain none or only one hidden layer. Therefore, these traditional machine learning models are also called shallow models.

The purpose of this survey is to classify and summarize the machine learning-based IDSs proposed to date, abstract the main ideas of applying machine learning to security domain problems, and analyze the current challenges and future developments. For this survey, we selected representative papers published from 2015 to 2019, which reflect the current progress. Several previous surveys [3–5] have classified research efforts by their applied machine learning algorithms. These surveys are primarily intended to introduce different machine learning algorithms applied to IDSs, which can be helpful to machine learning researchers. However, this type of taxonomic system emphasizes specific implementation technologies rather than cyber security domain problems. As a result, these surveys do not directly address how to resolve IDS domain problems using machine learning. For coping with this problem, we propose a new data-centered IDS taxonomy in this survey, and introduce the related studies following this taxonomy.

Data objects are the most basic elements in IDS. Data objects carry features related to attack behaviors. Feature types and feature extraction methods differ among different data elements, causing the most appropriate machine learning models to also differ. Therefore, this survey thoroughly analyzes the data processed in cyber security and classifies IDSs on the basis of data sources. This taxonomy presents a path involving data–feature–attack behavior–detection model, which is convenient for readers to find study ideas for particular domain problems. For example, this taxonomic system can answer the following problems: (1) What features best represent different attacks? (2) What type of data is most suitable for detecting certain attacks? (3) What types of machine learning algorithms are the best fit for a specific data type? (4) How do machine learning methods improve IDSs along different aspects? These problems appeal to cyber security researchers. Finally, the challenges and future development of machine learning methods for IDS are discussed by summarizing recent representative studies.

The rest of this paper is organized as follows: Section 2 introduces the key concepts and the taxonomy of IDS. Section 3 introduces the frequently used machine learning algorithms in IDS, their metrics, and common benchmark datasets. Section 4 classifies IDS according to data sources and sums up the process of applying machine learning to IDSs. Section 5 discusses the challenges and future directions of machine learning-based IDSs, and Section 6 concludes the paper.

## 2. Concept and Taxonomy of IDS

For an IDS, an intrusion means an attempt to access information about computer systems or to damage system operation in an illegal or unauthorized manner. An IDS is a computer-security application that aims to detect a wide range of security violations, ranging from attempted break-ins by outsiders to system penetrations and abuses by insiders [6]. The main functions of IDSs are to monitor hosts and networks, analyze the behaviors of computer systems, generate alerts, and respond to suspicious behaviors. Because they monitor related hosts and networks, IDSs are typically deployed near the protected network nodes (e.g., the switches in major network segments).

There are two types of IDS classification methods: detection-based method and data source-based methods. Among the detection-based methods, IDSs can be divided into misuse detection and

anomaly detection. Among the data source-based methods, IDSs can be divided into host-based and network-based methods [7]. This survey combines these two types of IDS classification methods, taking the data source as the main classification consideration and treating the detection method as a secondary classification element. The proposed taxonomy is shown in Figure 1. Regarding detection methods, the survey concentrates on machine learning methods. We introduce how to apply machine learning to IDS using different types of data in detail in Section 4.

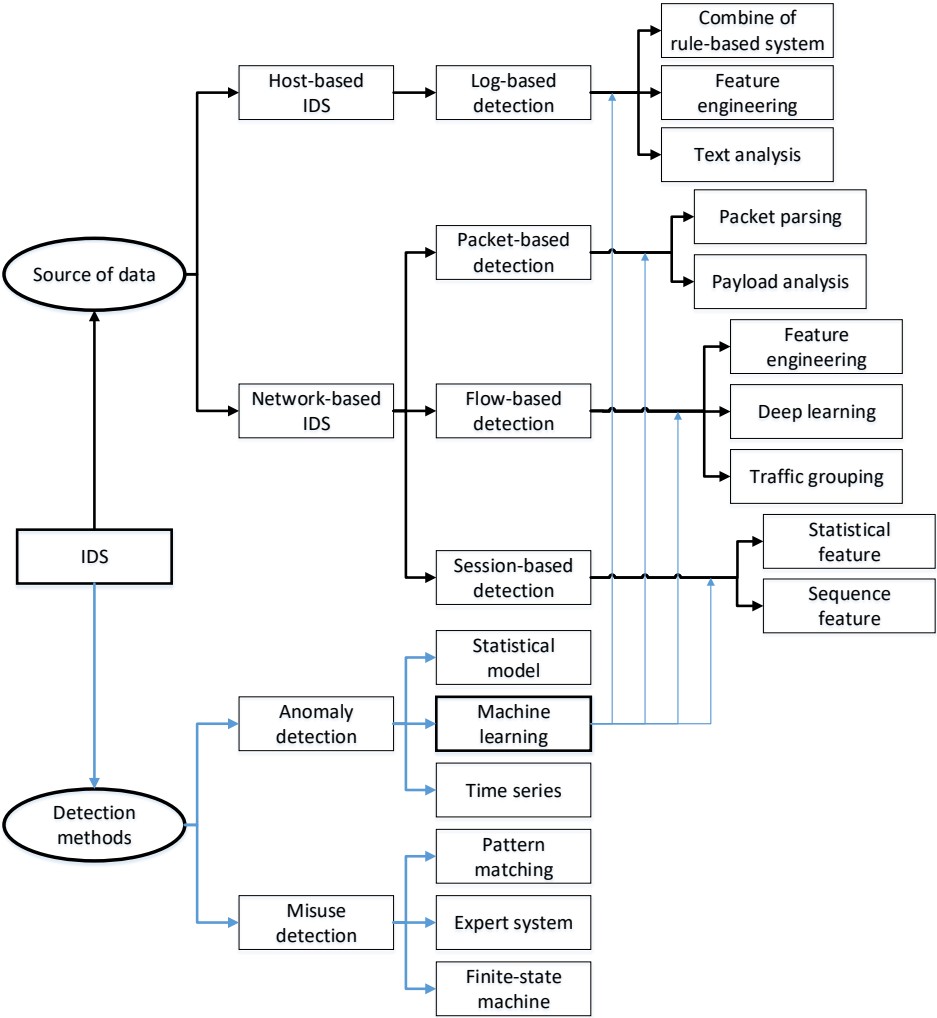

**Figure 1.** Taxonomy system of IDS.

## 2.1. Classification by Detection Methods

Misuse detection is also called signature-based detection. The basic idea to represent attack behaviors as signatures. The detection process matches the signatures of samples using a signature database. The main problem in constructing misuse detection systems is to design efficient signatures. The advantages of misuse detection are that it has a low false alarm rate and it reports attack types as well as possible reasons in detail; the disadvantages are that it has a high missed alarm rate, lacks the ability to detect unknown attacks, and requires maintaining a huge signature database. The design idea behind anomaly detection is to establish a normal behavior profile and then define abnormal behaviors by their degree of deviation from the normal profile. Thus, the key to designing an anomaly detection system is to clearly define a normal profile. The benefits of anomaly detection are strong generalizability and the ability to recognize unknown attacks, while its shortcomings are a high false alarm rate and an inability to provide possible reasons for an abnormality. The main differences between misuse detection and anomaly detection are listed in Table 1.

**Table 1.** Differences between misuse detection and anomaly detection.

|  | **Misuse Detection** | **Anomaly Detection** |
|---|---|---|
| **Detection performance** | Low false alarm rate; High missed alarm rate | Low missed alarm rate; High false alarm rate |
| **Detection efficiency** | High, decrease with scale of signature database | Dependent on model complexity |
| **Dependence on domain knowledge** | Almost all detections depend on domain knowledge | Low, only the feature design depends on domain knowledge |
| **Interpretation** | Design based on domain knowledge, strong interpretative ability | Outputs only detection results, weak interpretative ability |
| **Unknown attack detection** | Only detects known attacks | Detects known and unknown attacks |

As shown in Figure 1, in detection method-based taxonomy, misuse detection includes pattern matching-based, expert system, and finite state machine-based methods. Anomaly detection includes statistical model-based, machine learning-based, and time series-based methods.

### 2.2. Classification by Source of Data

An advantage of a host-based IDSs is that it can locate intrusions precisely and initiate responses because such IDSs can monitor the behaviors of significant objects (e.g., sensitive files, programs and ports). The disadvantages are that host-based IDSs occupy host resources, are dependent on the reliability of the host, and are unable to detect network attacks. A network-based IDS is usually deployed on major hosts or switches. A majority of network-based IDSs are independent of the operating system (OS); thus, they can be applied in different OS environments. Furthermore, network-based IDSs are able to detect specific types of protocol and network attacks. The drawback is that they monitor only the traffic passing through a specific network segment. The main differences between host-based IDS and network-based IDS are listed in Table 2.

**Table 2.** Differences between host-based and network-based IDSs.

|  | **Host-Based IDS** | **Network-Based IDS** |
|---|---|---|
| **Source of data** | Logs of operating system or application programs | Network traffic |
| **Deployment** | Every host; Dependent on operating systems; Difficult to deploy | Key network nodes; Easy to deploy |
| **Detection efficiency** | Low, must process numerous logs | High, can detect attacks in real time |
| **Intrusion traceability** | Trace the process of intrusion according to system call paths | Trace position and time of intrusion according to IP addresses and timestamps |
| **Limitation** | Cannot analyze network behaviors | Monitor only the traffic passing through a specific network segment |

As shown in Figure 1, a host-based IDS uses audit logs as a data source. Log detection methods are mainly hybrids based on rule and machine learning, rely on log features, and use text analysis-based methods. A network-based IDS uses network traffic as a data source—typically packets, which are the basic units of network communication. A flow is the set of packets within a time window, which reflects the network environment. A session is a packet sequence combined on the basis of a network information 5-tuple (client IP, client port, server IP, server port, protocol). A session represents high-level semantic information of traffic. Packets contain packet headers and payloads; therefore, packet detection includes parsing-based and payload analysis-based methods. Based on feature

extraction, flow detection can be divided into feature engineering-based and deep learning-based methods. In addition, traffic grouping is a unique approach in flow detection. Based on whether sequence information is used, session detection can be divided into statistical feature-based and sequence feature-based methods.

## 3. Common Machine Learning Algorithms in IDS

### 3.1. Machine Learning Models

There are two main types of machine learning: supervised and unsupervised learning. Supervised learning relies on useful information in labeled data. Classification is the most common task in supervised learning (and is also used most frequently in IDS); however, labeling data manually is expensive and time consuming. Consequently, the lack of sufficient labeled data forms the main bottleneck to supervised learning. In contrast, unsupervised learning extracts valuable feature information from unlabeled data, making it much easier to obtain training data. However, the detection performance of unsupervised learning methods is usually inferior to those of supervised learning methods. The common machine learning algorithms used in IDSs are shown in Figure 2.

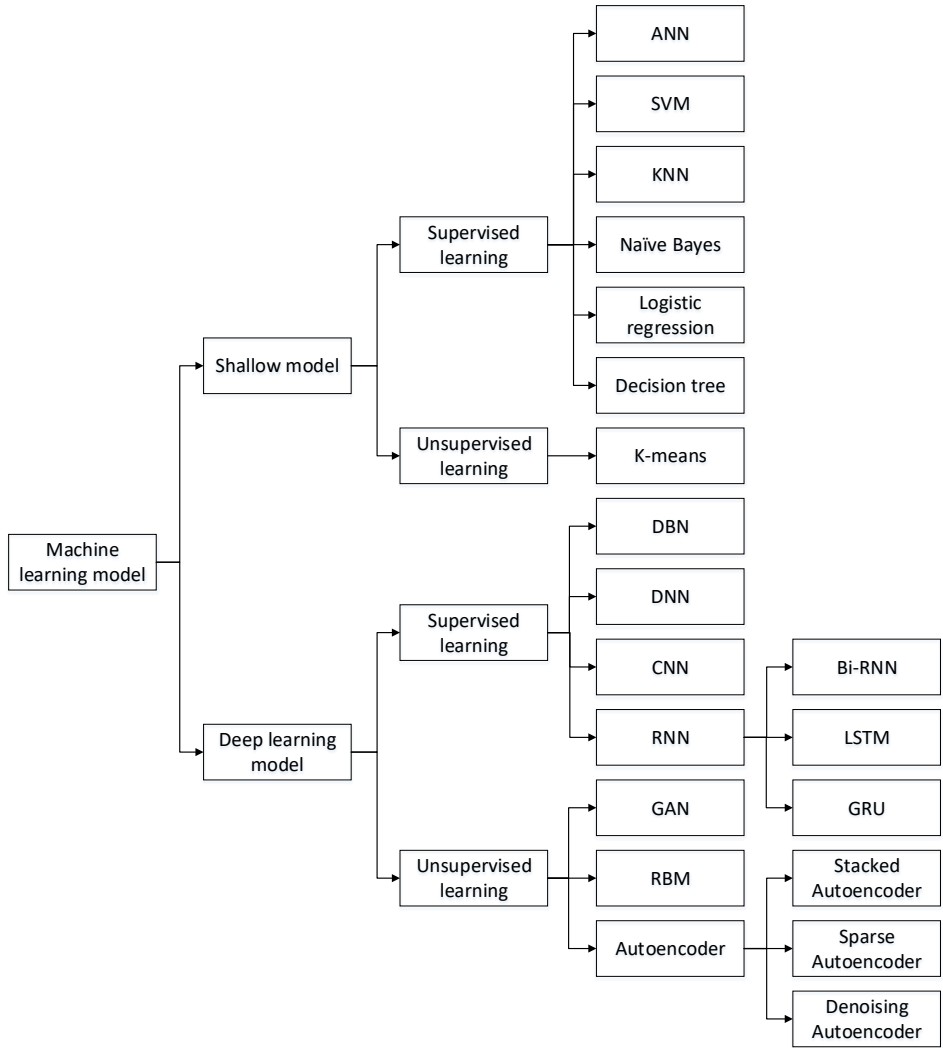

**Figure 2.** Taxonomy of machine learning algorithms.

### 3.1.1. Shallow Models

The traditional machine learning models (shallow models) for IDS primarily include the artificial neural network (ANN), support vector machine (SVM), K-nearest neighbor (KNN), naïve Bayes, logistic regression (LR), decision tree, clustering, and combined and hybrid methods. Some of these methods have been studied for several decades, and their methodology is mature. They focus not only on the detection effect but also on practical problems, e.g., detection efficiency and data management. The pros and cons of various shallow models are shown in Table 3.

**Table 3.** The pros and cons of various shallow models.

| Algorithms | Advantages | Disadvantages | Improvement Measures |
| --- | --- | --- | --- |
| ANN | Able to deal with nonlinear data; Strong fitting ability | Apt to overfitting; Prone to become stuck in a local optimum; Model training is time consuming | Adopted improved optimizers, activation functions, and loss functions |
| SVM | Learn useful information from small train set; Strong generation capability | Do not perform well on big data or multiple classification tasks; Sensitive to kernel function parameters | Optimized parameters by particle swarm optimization (PSO)[8] |
| KNN | Apply to massive data; Suitable to nonlinear data; Train quickly; Robust to noise | Low accuracy on the minority class; Long test times; Sensitive to the parameter $K$ | Reduced comparison times by trigonometric inequality; Optimized parameters by particle swarm optimization (PSO) [9]; Balanced datasets using the synthetic minority oversampling technique (SMOTE) [10] |
| Naïve Bayes | Robust to noise; Able to learn incrementally | Do not perform well on attribute-related data | Imported latent variables to relax the independent assumption [11] |
| LR | Simple, can be trained rapidly; Automatically scale features | Do not perform well on nonlinear data; Apt to overfitting | Imported regularization to avoid overfitting [12] |
| Decision tree | Automatically select features; Strong interpretation | Classification result trends to majority class; Ignore the correlation of data | Balanced datasets with SMOTE; Introduced latent variables |
| K-means | Simple, can be trained rapidly; Strong scalability; Can fit to big data | Do not perform well on nonconvex data; Sensitive to initialization; Sensitive to the parameter $K$ | Improved initialization method [13] |

**Artificial Neural Network (ANN)**. The design idea of an ANN is to mimic the way human brains work. An ANN contains an input layer, several hidden layers, and an output layer. The units in adjacent layers are fully connected. An ANN contains a huge number of units and can theoretically approximate arbitrary functions; hence, it has strong fitting ability, especially for nonlinear functions. Due to the complex model structure, training ANNs is time-consuming. It is noteworthy that ANN models are trained by the backpropagation algorithm that cannot be used to train deep networks. Thus, an ANN belongs to shallow models and differs from the deep learning models discussed in Section 3.1.2.

**Support Vector Machine (SVM)**. The strategy in SVMs is to find a max-margin separation hyperplane in the n-dimension feature space. SVMs can achieve gratifying results even with small-scale training sets because the separation hyperplane is determined only by a small number of support vectors. However, SVMs are sensitive to noise near the hyperplane. SVMs are able to solve linear

problems well. For nonlinear data, kernel functions are usually used. A kernel function maps the original space into a new space so that the original nonlinear data can be separated. Kernel tricks are widespread among both SVMs and other machine learning algorithms.

**K-Nearest Neighbor (KNN)**. The core idea of KNN is based on the manifold hypothesis. If most of a sample's neighbors belong to the same class, the sample has a high probability of belonging to the class. Thus, the classification result is only related to the top-k nearest neighbors. The parameter *k* greatly influences the performance of KNN models. The smaller k is, the more complex the model is and the higher the risk of overfitting. Conversely, the larger *k* is, the simpler the model is and the weaker the fitting ability.

**Naïve Bayes**. The Naïve Bayes algorithm is based on the conditional probability and the hypothesis of attribute independence. For every sample, the Naïve Bayes classifier calculates the conditional probabilities for different classes. The sample is classified into the maximum probability class. The conditional probability formula is calculated as shown in Formula (1).

$$P(X = x | Y = c_k) = \prod_{i=1}^{n} P(X^{(i)} = x^{(i)} | Y = c_k) \tag{1}$$

When the attribute independence hypothesis is satisfied, the Naïve Bayes algorithm reaches the optimal result. Unfortunately, that hypothesis is difficult to satisfy in reality; hence, the Naïve Bayes algorithm does not perform well on attribute-related data.

**Logistic Regression (LR)**. The LR is a type of logarithm linear model. The LR algorithm computes the probabilities of different classes through parametric logistic distribution, calculated as shown in Formula (2).

$$P(Y = k | x) = \frac{e^{w_k * x}}{1 + \sum_{k}^{K-1} e^{w_k * x}} \tag{2}$$

where *k = 1,2...K − 1*. The sample *x* is classified into the maximum probability class. An LR model is easy to construct, and model training is efficient. However, LR cannot deal well with nonlinear data, which limits its application.

**Decision tree**. The decision tree algorithm classifies data using a series of rules. The model is tree like, which makes it interpretable. The decision tree algorithm can automatically exclude irrelevant and redundant features. The learning process includes feature selection, tree generation, and tree pruning. When training a decision tree model, the algorithm selects the most suitable features individually and generates child nodes from the root node. The decision tree is a basic classifier. Some advanced algorithms, such as the random forest and the extreme gradient boosting (XGBoost), consist of multiple decision trees.

**Clustering**. Clustering is based on similarity theory, i.e., grouping highly similar data into the same clusters and grouping less-similar data into different clusters. Different from classification, clustering is a type of unsupervised learning. No prior knowledge or labeled data is needed for clustering algorithms; therefore, the data set requirements are relatively low. However, when using clustering algorithms to detect attacks, it is necessary to refer external information.

K-means is a typical clustering algorithm, where K is the number of clusters and the means is the mean of attributes. The K-means algorithm uses distance as a similarity measure criterion. The shorter the distance between two data objects is, the more likely they are to be placed in the same cluster. The K-means algorithm adapts well to linear data, but its results on nonconvex data are not ideal. In addition, the K-means algorithm is sensitive to the initialization condition and the parameter *K*. Consequently, many repeated experiments must be run to set the proper parameter value.

**Ensembles and Hybrids**. Every individual classifier has strengths and shortcomings. A natural approach is to combine various weak classifiers to implement a strong classifier. Ensemble methods train multiple classifiers; then, the classifiers vote to obtain the final results. Hybrid methods are designed as many stages, in which each stage uses a classification model. Because ensemble and hybrid

classifiers usually perform better than do single classifiers, an increasing number of researchers have begun to study ensemble and hybrid classifiers. The key points lie in selecting which classifiers to combine and how they are combined.

### 3.1.2. Deep Learning Models

Deep learning models consist of diverse deep networks. Among them, deep brief networks (DBNs), deep neural networks (DNNs), convolutional neural networks (CNNs), and recurrent neural networks (RNNs) are supervised learning models, while autoencoders, restricted Boltzmann machines (RBMs), and generative adversarial networks (GANs) are unsupervised learning models. The number of studies of deep learning-based IDSs has increased rapidly from 2015 to the present. Deep learning models directly learn feature representations from the original data, such as images and texts, without requiring manual feature engineering. Thus, deep learning methods can execute in an end-to-end manner. For large datasets, deep learning methods have a significant advantage over shallow models. In the study of deep learning, the main emphases are network architecture, hyperparameter selection, and optimization strategy. A comparison of various deep learning models is shown in Table 4.

**Table 4.** Comparison of various deep learning models

| Algorithms | Suitable Data Types | Supervised or Unsupervised | Functions |
|---|---|---|---|
| **Autoencoder** | Raw data; Feature vectors | Unsupervised | Feature extraction; Feature reduction; Denoising |
| **RBM** | Feature vectors | Unsupervised | Feature extraction; Feature reduction; Denoising |
| **DBN** | Feature vectors | Supervised | Feature extraction; Classification |
| **DNN** | Feature vectors | Supervised | Feature extraction; Classification |
| **CNN** | Raw data; Feature vectors; Matrices | Supervised | Feature extraction; Classification |
| **RNN** | Raw data; Feature vectors; Sequence data | Supervised | Feature extraction; Classification |
| **GAN** | Raw data; Feature vectors | Unsupervised | Data augmentation; Adversarial training |

**Autoencoder**. An autoencoder contains two symmetrical components, an encoder and a decoder, as shown in Figure 3. The encoder extracts features from raw data, and the decoder reconstructs the data from the extracted features. During training, the divergence between the input of the encoder and the output of the decoder is gradually reduced. When the decoder succeeds in reconstructing the data via the extracted features, it means that the features extracted by the encoder represent the essence of the data. It is important to note that this entire process requires no supervised information. Many famous autoencoder variants exist, such as denoising autoencoders [14,15] and sparse autoencoders [16].

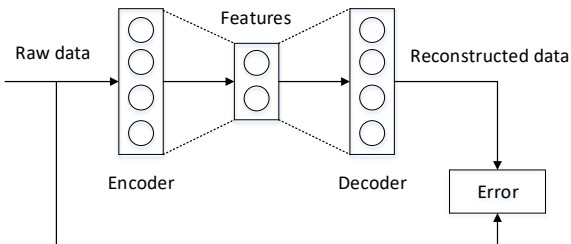

**Figure 3.** The structure of an autoencoder.

**Restricted Boltzmann Machine (RBM)**. An RBM is a randomized neural network in which units obey the Boltzmann distribution. An RBM is composed of a visible layer and a hidden layer. The units in the same layer are not connected; however, the units in different layers are fully connected, as shown in Figure 4. where $v_i$ is a visible layer, and $h_i$ is a hidden layer. RBMs do not distinguish between the forward and backward directions; thus, the weights in both directions are the same. RBMs are unsupervised learning models trained by the contrastive divergence algorithm [17], and they are usually applied for feature extraction or denoising.

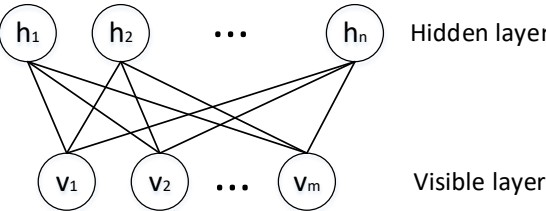

**Figure 4.** The structure of the RBM.

**Deep Brief Network (DBN)**. A DBN consists of several RBM layers and a softmax classification layer, as shown in Figure 5. Training a DBN involves two stages: unsupervised pretraining and supervised fine-tuning [18,19]. First, each RBM is trained using greedy layer-wise pretraining. Then, the weight of the softmax layer are learned by labeled data. In attack detection, DBNs are used for both feature extraction and classification [20–22].

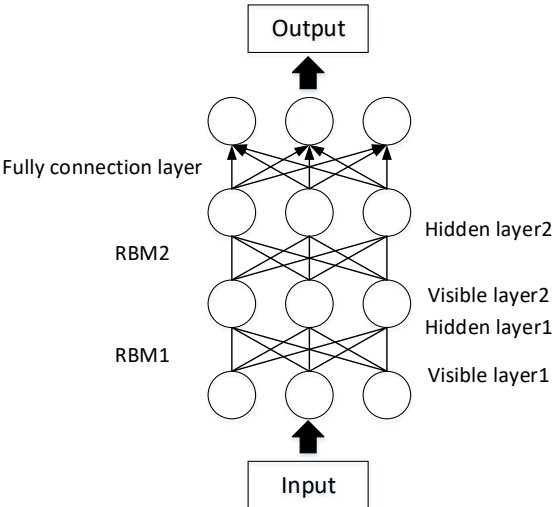

**Figure 5.** The structure of the DBN.

**Deep Neural Network (DNN)**. A layer-wise pretraining and fine-tuning strategy makes it possible to construct DNNs with multiple layers, as shown in Figure 6. When training a DNN,

the parameters are learned first using unlabeled data, which is an unsupervised feature learning stage; then, the network is tuned through the labeled data, which is a supervised learning stage. The astonishing achievements of DNNs are mainly due to the unsupervised feature learning stage.

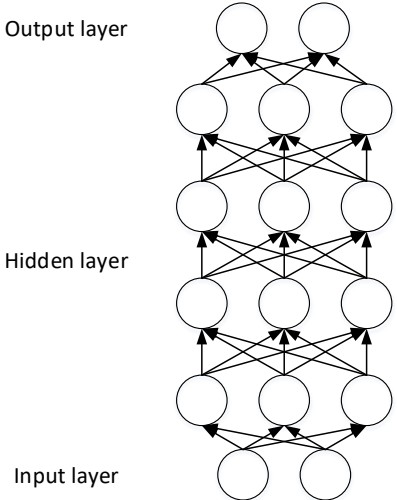

**Figure 6.** The structure of the DNN.

**Convolutional Neural Network (CNN)**. CNNs are designed to mimic the human visual system (HVS); consequently, CNNs have made great achievements in the computer vision field [23–25]. A CNN is stacked with alternate convolutional and pooling layers, as shown in Figure 7. The convolutional layers are used to extract features, and the pooling layers are used to enhance the feature generalizability. CNNs work on 2-dimensional (2D) data, so the input data must be translated into matrices for attack detection.

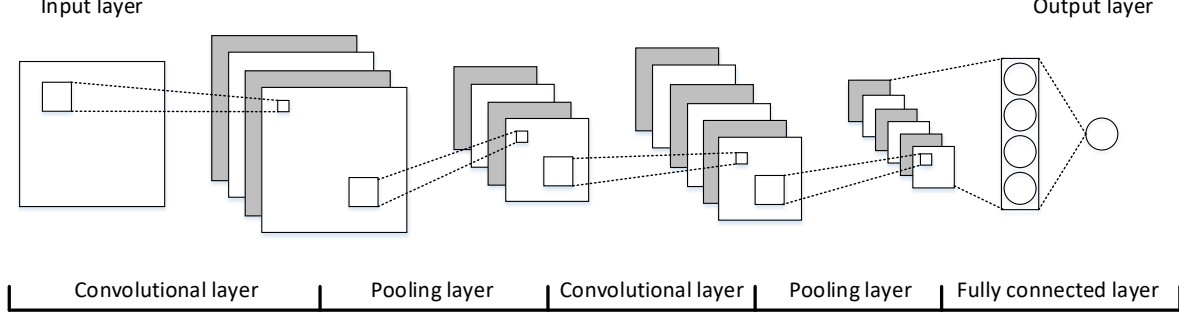

**Figure 7.** The structure of a CNN.

**Recurrent Neural Network (RNN)**. RNNs are networks designed for sequential data and are widely used in natural language processing (NLP) [26–28]. The characteristics of sequential data are contextual; analyzing isolated data from the sequence makes no sense. To obtain contextual information, each unit in an RNN receives not only the current state but also previous states. The structure of an RNN is shown in Figure 8. Where all the *W* items in Figure 8 are the same. This characteristic causes RNNs to often suffer from vanishing or exploding gradients. In reality, standard RNNs deal with only limited-length sequences. To solve the long-term dependence problem, many RNN variants have been proposed, such as long short-term memory (LSTM) [29], gated recurrent unit (GRU) [30], and bi-RNN [31].

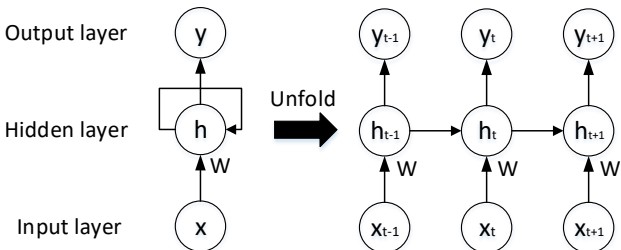

**Figure 8.** The structure of an RNN.

The LSTM model was proposed by Hochreiter and Schmidhuber in 1997 [29]. Each LSTM unit contains three gates: a forget gate, an input gate, and an output gate. The forget gate eliminates outdated memory, the input gate receives new data, and the output gate combines short-term memory with long-term memory to generate the current memory state. The GRU was proposed by Chung et al. in 2014 [30]. The GRU model merges the forget gate and the input gate into a single update gate, which is simpler than the LSTM.

**Generative Adversarial Network (GAN)**. A GAN model includes two subnetworks, i.e., a generator and a discriminator. The generator aims to generate synthetic data similar to the real data, and the discriminator intends to distinguish synthetic data from real data. Thus, the generator and the discriminator improve each other. GANs are currently a hot research topic used to augment data in attack detection, which partly ease the problem of IDS dataset shortages. Meanwhile, GANs belong to adversarial learning approaches which can raise the detection accuracy of models by adding adversarial samples to the training set.

### 3.1.3. Shallow Models Compared to Deep Models

Deep learning is a branch of machine learning, and the effects of deep learning models are obviously superior to those of the traditional machine learning (or shallow model) methods in most application scenarios. The differences between shallow models and deep models are mainly reflected in the following aspects.

**(1) Running time**. The running time includes both training and test time. Due to the high complexity of deep models, both their training and test times are much longer than those of shallow models.

**(2) Number of parameters**. There are two types of parameters: learnable parameters and hyperparameters. The learnable parameters are calculated during the training phase, and the hyperparameters are set manually before training begins. The learnable parameters and hyperparameters in deep models far outnumber those in shallow models; consequently, training and optimizing deep models takes longer.

**(3) Feature representation**. The input to traditional machine learning models is a feature vector, and feature engineering is an essential step. In contrast, deep learning models are able to learn feature representations from raw data and are not reliant on feature engineering. The deep learning methods can execute in an end-to-end manner, giving them an outstanding advantage over traditional machine learning methods.

**(4) Learning capacity**. The structures of deep learning models are complex and they contain huge numbers of parameters (generally millions or more). Therefore, the deep learning models have stronger fitting ability than do shallow models. However, deep learning models also face a higher risk of overfitting, require a much larger volume of data for training. However, the effect of deep learning models is better.

**(5) Interpretability**. Deep learning models are black boxes [32–35]; the results are almost uninterpretable, which is a critical point in deep learning. However, some traditional deep learning algorithms, such as the decision tree and naïve Bayes, have strong interpretability.

*3.2. Metrics*

Many metrics are used to evaluate machine learning methods. The optimal models are selected using these metrics. To comprehensively measure the detection effect, multiple metrics are often used simultaneously in IDS research.

- **Accuracy** is defined as the ratio of correctly classified samples to total samples. Accuracy is a suitable metric when the dataset is balanced. In real network environments; however, normal samples are far more abundant than are abnormal samples; thus, accuracy may not be a suitable metric.

$$Accuracy = \frac{TP + TN}{TP + FP + FN + TN} \tag{3}$$

- **Precision (P)** is defined as the ratio of true positive samples to predicted positive samples; it represents the confidence of attack detection.

$$P = \frac{TP}{TP + FP} \tag{4}$$

- **Recall (R)** is defined as the ratio of true positive samples to total positive samples and is also called the detection rate. The detection rate reflects the model's ability to recognize attacks, which is an important metric in IDS.

$$R = \frac{TP}{TP + FN} \tag{5}$$

- **F-measure (F)** is defined as the harmonic average of the precision and the recall.

$$F = \frac{2 * P * R}{P + R} \tag{6}$$

- **The false negative rate (FNR)** is defined as the ratio of false negative samples to total positive samples. In attack detection, the FNR is also called the missed alarm rate.

$$FNR = \frac{FN}{TP + FN} \tag{7}$$

- **The false positive rate (FPR)** is defined as the ratio of false positive samples to predicted positive samples. In attack detection, the FPR is also called the false alarm rate, and it is calculated as follows:

$$FPR = \frac{FP}{TP + FP} \tag{8}$$

where the TP is the true positives, FP is the false positives, TN is the true negatives, FN is the false negatives. The purpose of an IDS is to recognize attacks; therefore, attack samples are usually regarded as positives, and normal samples are usually regarded as negatives. In attack detection, the frequently used metrics include accuracy, recall (or detection rate), FNR (or missed alarm rate), and FPR (or false alarm rate).

*3.3. Benchmark Datasets in IDS*

The task of machine learning is to extract valuable information from data; therefore, the performance of machine learning depends upon the quality of the input data. Understanding data is the basis of machine learning methodology. For IDSs, the adopted data should be easy to acquire and reflect the behaviors of the hosts or networks. The common source data types for IDSs are packets, flow, sessions, and logs. Building a dataset is complex and time-consuming. After a benchmark dataset is constructed, it can be reused repeatedly by many researchers. In addition to convenience, there are two other benefits of using benchmark datasets. (1) The benchmark datasets are authoritative, and make experimental results more convincing. (2) Many published studies have been conducted

using common benchmark datasets, which allows new study results to be compared with those of previous studies.

(1)   DARPA1998

The DARPA1998 dataset [36] was built by the Lincoln laboratory of MIT and is a widely used benchmark dataset in IDS studies. To compile it, the researchers collected Internet traffic over nine weeks; the first seven weeks form the training set, and the last two weeks form the test set. The dataset contains both raw packets and labels. There are five types of labels: *normal*, *denial of service (DOS)*, *Probe*, *User to Root (U2R)* and *Remote to Local (R2L)*. Because raw packets cannot be directly applied to traditional machine learning models, the KDD99 dataset was constructed to overcome this drawback.

(2)   KDD99

The KDD99 [37] dataset is the most widespread IDS benchmark dataset at present. Its compilers extracted 41-dimensional features from data in DARPA1998. The labels in KDD99 are the same as the DARPA1998. There are four types of features in KDD99, i.e., *basic features*, *content features*, *host-based statistical features*, and *time-based statistical features*. Unfortunately, the KDD99 dataset includes many defects. First, the data are severely unbalanced, making the classification results biased toward the majority classes. Additionally, there are many duplicate records and redundant records exist. Many researchers have to filter the dataset carefully before they can use it. As a result, the experimental results from different studies are not always comparable. Last but not least, KDD data are too old to represent the current network environment.

(3)   NSL-KDD

To overcome the shortcomings of the KDD99 dataset, the NSL-KDD [38] was proposed. The records in the NSL-KDD were carefully selected based on the KDD99. Records of different classes are balanced in the NSL-KDD, which avoids the classification bias problem. The NSL-KDD also removed duplicate and redundant records; therefore, it contains only a moderate number of records. Therefore, the experiments can be implemented on the whole dataset, and the results from different papers are consistent and comparable. The NSL-KDD alleviates the problems of data bias and data redundancy to some degree. However, the NSL-KDD does not include new data; thus, minority class samples are still lacking, and its samples are still out-of-date.

(4)   UNSW-NB15

The UNSW-NB15 [39] dataset was compiled by the University of South Wales, where researchers configured three virtual servers to capture network traffic and extracted 49-dimensional features using tool named Bro. The dataset includes more types of attacks than does the KDD99 dataset, and its features are more plentiful. The data categories include normal data and nine types of attacks. The features include *flow features*, *basic features*, *content features*, *time features*, *additional features*, and *labeled features*. The UNSW-NB15 is representative of new IDS datasets, and has been used in some recent studies. Although the influence of UNSW-NB15 is currently inferior to that of KDD99, it is necessary to construct new datasets for developing new IDS based on machine learning.

## 4. Research on Machine Learning-Based IDSs

Machine learning is a type of data driven method in which understanding the data is the first step. Thus, we adopt the type of data source of as the main classification thread, as shown in Figure 1. In this section, we introduce various ways to apply machine learning to IDS design for different data types. The different types of data reflect different attack behaviors, which include host behaviors and network behaviors. Host behaviors are reflected by system logs, and network behaviors are reflected by network traffic. There are multiple attack types, each of which has a unique pattern. Thus, selecting appropriate data sources is required to detect different attacks according to the attack characteristics. For instance, one salient feature of a DOS attack is to send many packets within a very short period of

time; therefore flow data is suitable for detecting a DOS attack. A covert channel involves data-leaking activity between two specific IP addresses, which is more suited to detection from session data.

### 4.1. Packet-Based Attack Detection

Packets, which are the basic units of network communication, represent the details of each communication. Packets consist of binary data, meaning that they are incomprehensible unless they are first parsed. A packet consists of a header and application data. The headers are structured fields that specify IP addresses, ports and other fields specific to various protocols. The application data portion contains the payload from the application layer protocols. There are three advantages to using packets as IDS data sources: (1) Packets contain communication contents; thus, they can effectively be used to detect U2L and R2L attacks. (2) Packets contain IPs and timestamps; thus, they can locate the attack sources precisely. (3) Packets can be processed instantly without caching; thus, detection can occur in real time. However, individual packets do not reflect the full communication state nor the contextual information of each packet, so it is difficult to detect some attacks, such as DDOS. The detection methods based on packets mainly include packet parsing methods and payload analysis methods.

#### 4.1.1. Packet Parsing-Based Detection

Various types of protocols are used in network communications, such as HTTP and DNS. These protocols have different formats; the packet parsing-based detection methods primarily focus on the protocol header fields. The usual practice is to extract the header fields using parsing tools (such as Wireshark or the Bro) and then to treat the values of the most important fields as feature vectors. Packet parsing-based detection methods apply to shallow models.

The header fields provide basic packet information from which feature can be extracted used with using classification algorithms to detect attacks. Mayhew et al. [40] proposed an SVM- and K-means-based packet detection method. They captured packets from a real enterprise network and parsed them with Bro. First, they grouped the packets according to protocol type. Then, they clustered the data with the K-means++ algorithm for the different protocol datasets. Thus, the original dataset was grouped into many clusters, where the data from any given cluster were homologous. Next, they extracted features from the packets and trained SVM models on each cluster. Their precision scores for HTTP, TCP, Wiki, Twitter, and E-mail protocols reached 99.6%, 92.9%, 99%, 96%, and 93%, respectively.

In packet parsing-based detection, unsupervised learning is a common way to solve the high false alarm rate problem. Hu et al. [41] proposed a fuzzy C-means based packet detection method. The fuzzy C mean algorithm introduces fuzzy logic into the standard K-means algorithm such that samples belong to a cluster with a membership degree rather than as a Boolean value such as 0 or 1. They used Snort to process the DARPA 2000 dataset, extracting Snort alerts, source IPs, destination IPs, source ports, destination ports, and timestamps. Then, they used this information to form feature vectors and distinguished false alerts from true alerts by clustering the packets. To reduce the influence of initialization, they ran the clustering algorithms ten times. The results showed that the fuzzy C-means algorithm reduced the false alarm rate by 16.58% and the missed alarm rate by 19.23%.

#### 4.1.2. Payload Analysis-Based Detection

Apart from packet parsing-based detection, payload analysis-based detection places emphasis on the application data. The payload analysis-based methods are suitable for multiple protocols because they do not need to parse the packet headers.

As a type of unstructured data, payloads can be processed directly by deep learning models [42]. It should be noted that this method does not include encrypted payloads. Shallow models depend on manual features and private information in packets, leading to high labor costs and privacy leakage problems. Deep learning methods learn features from raw data without manual intervention. Min et al. [43] utilized a text-based CNN to detect attacks from payloads. They conducted

experiments on the ISCX 2012 dataset and detected attacks with both statistical and content features. The statistical features mainly came from packet headers and included protocols, IPs, and ports. The content features came from the payloads. First, payloads from different packets were concatenated. Next, the concatenated payloads were encoded by skip-gram word embedding. Then, the content features were extracted with a CNN. Finally, they trained a random forest model to detect attacks. The final model reached an accuracy of 99.13%.

Combining various payload analysis techniques can achieve comprehensive content information, which is able to improve the effect of the IDS. Zeng et al. [44] proposed a payload detection method with multiple deep learning models. They adopted three deep learning models (a CNN, an LSTM, and a stacked autoencoder) to extract features from different points of view. Among these, the CNN extracted local features, the RNN extracted time series features, and the stacked autoencoder extracted text features. The accuracy of this combined approach reached 99.22% on the ISCX 2012 dataset.

Extracting payload features with unsupervised learning is also an effective detection method. Yu et al. [45] utilized a convolutional autoencoder to extract payload features and conducted experiments on the CTU-UNB dataset. This dataset includes the raw packets of 8 attack types. To take full advantage of convolutions, they first converted the packets into images. Then, they trained a convolutional autoencoder model to extract features. Finally, they classified packets using learned features. The precision, recall and F-measure on the test set reached 98.44%, 98.40%, and 98.41% respectively.

To enhance the robustness of IDSs, adversarial learning becomes a novel approach. Adversarial learning can be used for attacks against IDS. Meanwhile, it is also a novel way to improve detection accuracy of IDS. Rigaki et al. [46] used a GAN to improve the malware detection effect. To evade detection, malware applications try to generate packets similar to normal packets. Taking the malware FLU as an example, the command & control (C & C) packets are very similar to packets generated by Facebook. They configured a virtual network system with hosts, servers, and an IPS. Then, they started up the malware FLU and trained a GAN model. The GAN guided the malware to produce packets similar to Facebook. As the training epochs increased, the packets blocked by the IPS decreased and packet that passed inspection increased. The result was that the malicious packets generated by the GAN were more similar to normal packets. Then, by analyzing the generated packets, the robustness of the IPS was improved.

### 4.2. Flow-Based Attack Detection

Flow data contains packets grouped in a period, which is the most widespread data source for IDSs. The KDD99 and the NSL-KDD datasets are both flow data. Detecting attacks with flow has two benefits: (1) Flow represents the whole network environment, which can detect most attacks, especially DOS and Probe. (2) Without packet parsing or session restructuring, flow preprocessing is simple. However, flow ignores the content of packets; thus, its detection effect for U2R and R2L is unsatisfactory. When extracting flow features, packets must be cached packets; thus, it involves some hysteresis. Flow-based attack detection mainly includes feature engineering and deep learning methods. In addition, the strong heterogeneity of flow may cause poor detection effects. Traffic grouping is the usual solution to this problem.

### 4.2.1. Feature Engineering-Based Detection

Traditional machine learning models cannot directly address flow data; therefore, feature engineering is an essential step before these models can be applied. Feature engineering-based methods adopt a "feature vectors + shallow models" mode. The feature vectors are suitable for most machine learning algorithms. Each dimension of the feature vectors has clear interpretable semantics. The common features include the average packet length, the variance in packet length, the ratio of TCP to UDP, the proportion of TCP flags, and so on. The advantages of these types of detection methods are that they are simple to implement, highly efficient, and can meet real-time requirements.

The existing feature engineering-based IDSs often have high detection accuracy but suffer from a high false alarm rate. One solution is to combine many weak classifiers to obtain a strong classifier. Goeschel et al. [47] proposed a hybrid method that included SVM, decision tree, and Naïve Bayes algorithms. They first trained an SVM model to divide the data into normal or abnormal samples. For the abnormal samples, they utilized a decision tree model to determine specific attack types. However, a decision tree model can identify only known attacks, not unknown attacks. Thus, they also applied a Naïve Bayes classifier to discover unknown attacks. By taking advantage of three different classifier types this hybrid method achieved an accuracy of 99.62% and a false alarm rate of 1.57% on the KDD99 dataset.

Another research objective is to accelerate the detection speed. Kuttranont et al. [48] proposed a KNN-based detection method and accelerated calculation via parallel computing techniques running on a graphics processing unit (GPU). They modified the neighbor-selecting rule of the KNN algorithm. The standard KNN selects the top K nearest samples as neighbors, while the improved algorithm selects a fixed percentage (such as 50%) of the neighboring samples as neighbors. The proposed method considers the unevenness of data distribution and performs well on sparse data. These experiments were conducted using the KDD99 dataset, achieving an accuracy of 99.30%. They also applied parallel computing and the GPU to accelerating calculation. The experimental results showed that the method with the GPU was approximately 30 times faster than that without the GPU.

The unsupervised learning methods are also applied to IDS, a typical way is to divide data with clustering algorithms. The standard K-means algorithm is inefficient on big datasets. To improve detection efficiency, Peng et al. [13] proposed an improved K-means detection method with mini batch. They first carried out data preprocessing on the KDD99 dataset. The nominal features were transformed into numerical types, and each dimension of the features was normalized by the max-min method. Then, they reduced the dimensions using the principal components analysis (PCA) algorithm. Finally, they clustered the samples with the K-means algorithm, but they improved K-means from two aspects. (1) They altered the method of initialization to avoid becoming stuck in a local optimum. (2) They introduced the mini-batch trick to decrease the running time. Compared with the standard K-means, the proposed method achieved higher accuracy and runtime efficiency.

### 4.2.2. Deep Learning-Based Detection

Feature engineering depends on domain knowledge, and the quality of features often becomes a bottleneck of detection effects. Deep learning-based detection methods learn feature automatically. These types of methods work in an end-to-end fashion and are gradually becoming the mainstream approach in IDS studies.

Deep learning methods can directly process raw data, allowing them to learn features and perform classification at the same time. Potluri et al. [49] proposed a CNN-based detection method. They conducted experiments on the NSL-KDD and the UNSW-NB 15 datasets. The data type in these datasets is a feature vector. Because CNNs are good at processing 2-dimensional (2D) data, they first converted the feature vectors into images. Nominal features were one-hot coded, and the feature dimensions increased from 41 to 464. Then, each 8-byte chunk was transformed into one pixel. Blank pixels were padded with 0. The end result was that the feature vectors were transformed into images of 8*8 pixels. Finally, they constructed a three-layer CNN to classify the attacks. They compared their model with other deep networks (ResNet 50 and GoogLeNet), and the proposed CNN performed best, reaching accuracies of 91.14% on the NSL-KDD and 94.9% on the UNSW-NB 15.

Unsupervised deep learning models can also be used to extract features; then, shallow models can be used to perform classification. Zhang et al. [50] extracted features with a sparse autoencoder and detected attacks with an XGBoost model. They used data from the NSL-KDD dataset. Due to the imbalanced nature of this dataset, they sampled the dataset using SMOTE. The SMOTE algorithm oversamples the minority classes and divides the majority classes into many subclasses so that every class is balanced. The sparse autoencoder introduces a sparsity constraint into the original autoencoder,

enhancing its ability to detect unknown samples. Finally, they classified the data using an XGBoost model. Their model achieved accuracies on the Normal, DOS, Probe, R2L, and U2R classes of 99.96%, 99.17%, 99.50%, 97.13%, and 89.00%, respectively.

Deep learning models have made great strides in big data analysis; however, their performances are not ideal on small or unbalanced datasets. Adversarial learning approaches can improve the detection accuracy on small datasets. Zhang et al. [51] conducted data augmentation with a GAN. The KDD99 dataset is both unbalanced and lacks new data, which leads to poor generalizability of machine learning models. To address these problems, they utilized a GAN to expand the dataset. The GAN model generated data similar to the flow data of KDD99. Adding this generated data to the training set allows attack variants to be detected. They selected 8 types of attacks and compared the accuracies achieved on the original dataset compared to the expanded dataset. The experimental results showed that adversarial learning improved 7 accuracies in 8 attack types.

### 4.2.3. Traffic Grouping-Based Detection

Flow includes all traffic within a period, and many types of traffics may act as white noise in attack detection. Training machine learning models with such data probably leads to overfitting. One natural approach is to group traffic to decrease heterogeneity. The grouping methods include protocol-based and data-based methods.

The traffic features of various protocols have significant differences; thus, grouping traffic by protocol is a valid step toward improving accuracy. Teng et al. [52] proposed an SVM detection method based on protocol grouping using the data of the KDD99 dataset, which involves various protocols. They first divided the dataset based on protocol type, and considered only TCP, UDP, and ICMP protocols. Then, according to the characteristics of these different protocols, they selected features for each subdataset. Finally, they trained SVM models on the 3 subdatasets, obtaining an average accuracy of 89.02%.

Grouping based on data characteristics is another traffic grouping approach. One typical method is clustering. Ma et al. [53] proposed a DNN and spectral clustering-based detection method. The heterogeneity of flow may cause low accuracy. Therefore, they first divided the original dataset into 6 subsets, in which each subset was highly homogeneous. Then, they trained DNN models on every subset. The accuracy of their approach on the KDD99 and the NSL-KDD datasets reached 92.1%.

### *4.3. Session-Based Attack Detection*

A session is the interaction process between two terminal applications and can represent high-level semantics. A session is usually divided on the basis of a 5-tuple (client IP, client port, server IP, server port, and protocol). There are two advantages of detection using sessions. (1) Sessions are suitable for detecting an attack between specific IP addresses, such as tunnel and Trojan attacks. (2) Sessions contain detailed communications between the attacker and the victim, which can help localize attack sources. However, session duration can vary dramatically. As a result, a session analysis sometimes needs to cache many packets, which may increase lag. The session-based detection methods primarily include statistics-based features and sequence-based features.

### 4.3.1. Statistic-Based Feature Detection Methods

Session statistical information includes the fields in packet headers, the number of packets, the proportion of packets coming from different directions, and so on. This statistical information is used to compose feature vectors suitable for shallow models. The sessions have high layer semantics; thus, they are easily described by rules. Decision tree or rule-based models may be appropriate methods. Unfortunately, the methods based on statistical features ignore the sequence information, and they have difficulties detecting intrusions related to communication content.

Because statistical information includes the basic features of sessions, supervised learning methods can utilize such information to differentiate between normal sessions and abnormal sessions.

The existing session-based detection methods often face problems of low accuracy and have high runtime costs. Ahmim et al. [54] proposed a hierarchical decision tree method in which, reduce the detection time, they analyzed the frequency of different types of attacks and designed the detection system to recognize specific attacks. They used data from the CICIDS 2017 dataset that included 79-dimensional features and 15 classes. The proposed detection system had a two-layer structure. The first layer consisted of two independent classifiers (i.e., a decision tree and a rule-based model), which processed part of the features. The second layer was a random forest classifier, which processed all the features from the dataset as well as the output of the first layer. They compared multiple machine learning models on 15 classes; their proposed methods performed best on 8 of the 15 classes. Moreover, the proposed method had low time consumption, reflecting its practicability.

Session-based detection using supervised learning models depends on expert knowledge, which is difficult to expand to new scenarios. To address this problem, Alseiari et al. [55] proposed an unsupervised method to detect attacks in smart grids. Due to the lack of smart grid datasets, they constructed a dataset through simulation experiments. First, they captured and cached packets to construct sessions. Then, they extracted 23-dimensional features from the sessions. Next, they utilized mini batch K-means to divide the data into many clusters. Finally, they labeled the clusters. This work was based on two hypotheses. The first was that normal samples were the majority. The second one was that the distances among the normal clusters were relatively short. When the size of a cluster was less than 25% of the full sample amount or a cluster centroid was far away from all other the other cluster centroids, that cluster was judged as abnormal. No expert knowledge was required for any part of this process. The proposed methods were able to detect intrusion behaviors in smart grids effectively and locate the attack sources while holding the false alarm rate less to than 5%.

### 4.3.2. Sequence Feature-Based Detection

Different from flow, the packets in sessions have a strict order relationship. The sequence features mainly contain the packet length sequence and the time interval sequence. Analyzing the sequence can obtain detailed session interaction information. Most machine learning algorithms cannot deal with sequences, and related methods are relatively rare. At present, most sequence feature-based detection adopts the RNN algorithm.

Encoding raw data is a common preprocessing step for RNN methods. The bag of words (BoW) model is a frequently used text processing technology. Yuan et al. [56] proposed a DDOS detection method based on the LSTM using UNB ISCX 2012 dataset. They first extracted 20-dimensional features from the packets and encoded them with BoW. Then, they concatenated the packets in sequence, resulting in matrices with a size of m*n, where m was the number of packets in a session and n was the dimension of a packet, and both m and n were variable. Finally, they trained a CNN to extract local features and an LSTM to classify the sessions. They provided comprehensive experimental results, reaching accuracy, precision, recall, and F-measure scores of 97.606%, 97.832%, 97.378%, and 97.601%, respectively.

One of the drawbacks of the BoW is that it is unable to represent the similarity between words. Word embedding approaches overcome that problem. Radford et al. [57] proposed a session detection method based on a bi-LSTM. Because LSTMs had made great strides in NLP, they expressed the sessions as a specific language. They conducted experiments on the ISCX IDS dataset. First, they grouped packets on the basis of IP addresses to obtain sessions. Then, they encoded the sessions with the word embedding. Finally, they trained an LSTM model to predict abnormal sessions. To utilize the contextual information, they adopted a bi-LSTM model to learn the sequence features in two directions.

In addition to text processing technology, the character-level CNN is a novel encoding method. Wang et al. [58] proposed a hierarchical deep learning detection method in which a session contains not only packet contents but also the packet time sequence. Then, they designed a hierarchical deep learning method using a CNN to learn the low-level spatial features and an LSTM to learn the high-level time features, where the time features are based on the spatial features. They conducted

experiments on the DARPA 1998 and the ISCX 2012 datasets. They first applied the CNN to extract spatial features from packets. Next, they concatenated the spatial features in sequence and extracted time features using the LSTM model. The resulting model achieved accuracies between 99.92% and 99.96%, and detection rates between 95.76% and 98.99%.

*4.4. Log-Based Attack Detection*

Logs are the activity records of operating systems or application programs; they include system calls, alert logs, and access records. Logs have definite semantics. There are three benefits to using logs as a data source in IDSs. (1) Logs include detailed content information suitable for detecting SQL injection, U2R, and R2L attacks. (2) Logs often carry information about users and timestamps that can be used to trace attackers and reveal attack times. (3) Logs record the complete intrusion process; thus, the result is interpretable. However, one problem is that log analysis depends on cyber security knowledge. Additionally, the log formats of different application programs do not have identical formats, resulting in low scalability. The log-based attack detection primarily includes hybrid methods involving rules and machine learning, log feature extraction-based methods, and text analysis-based methods.

4.4.1. Rule and Machine Learning-Based Hybrid Methods

Hybrid methods combine rule-based detection and machine learning, which together achieve better performances than do single detection systems. Many rule-based detection systems (e.g., Snort) generate masses of alerts; however, most of the alerts involve only operations that do not match the rules; therefore, these are often not real intrusion behaviors. The hybrid methods take the log output of the rule-based systems as inputs; then, machine learning models are used to filter out the meaningless alerts.

Many IDSs suffer from high false alarm rates, which cause real attacks to be embedded among many meaningless alerts. Ranking alerts via machine learning models forms a possible solution. To reduce the false alarm rate, Meng et al. [59] proposed a KNN method to filter alarms. They conducted experiments in a real network environment and generated alerts using Snort. Then, they trained a KNN model to rank the alerts. There were 5 threat levels in total in their experiment, and the results showed that the KNN model reduced the number of alerts by 89%.

Some IDSs perform a function similar to human interaction, in which alerts are ranked by machine learning to reduce analyst workloads. McElwee et al. [60] proposed an alert filtering method based on a DNN. They first collected the log generated by McAfee. Then, they trained a DNN model to find important security events in the logs. Next, the extracted important events were analyzed by security experts. Then, the analysis results were used as training data to enhance the DNN model, forming an interaction and promotion cycle. The proposed hybrid system can reduce analyst workloads and accelerate security analyses.

4.4.2. Log Feature Extraction-Based Detection

This method involves extracting log features according to domain knowledge and discovering abnormal behaviors using the extracted features, which is suitable for most machine learning algorithms. Using a sliding window to extract features is a common approach. The sliding window makes use of the contextual information contained in logs. In addition, the sliding window is a streaming method that has the benefit of low delay.

Intrusion behaviors may leave traces of system calls, and analyzing these system calls with classification algorithms can detect intrusions. Tran et al. [61] proposed a CNN method to analyze system calls. Every underlying operation that involves the operating system will use system calls; thus, analyzing the system call path can reproduce the complete intrusion process. They conducted experiments on the NGIDS-DS and the ADFA-LD datasets, which include a series of system calls. First, they extracted features with a sliding window. Then, they applied a CNN model to perform

classification. The CNN was good at finding local relationships and detecting abnormal behaviors from system calls.

Model interpretation is another important research direction, which has attracted extensive attention. Tuor et al. [62] proposed an interpretable deep learning detection method using data from the CERT Insider Threat dataset, which consists of system logs. They first extracted 414-dimensional features using a sliding window. Then, they adopted a DNN and an RNN to classify logs. The DNN detected attacks based on the log contents, and the RNN detected attacks based on the log sequences. The proposed methods reduced the analysis workload by 93.5% and reached a detection rate of 90%. Furthermore, they decomposed the abnormal scores into the contributions of each behavior, which was a helpful analysis. Interpretable models are more convincing than are uninterpretable models.

Some logs lack labeled information; consequently, supervised learning is inappropriate. Unsupervised learning methods are usually used with unlabeled logs. Bohara et al. [63] proposed an unsupervised learning detection method in the enterprise environment. They conducted experiments on the VAST 2011 Mini Challenge 2 dataset and extracted features from the host and network logs. Due to the different influences of each feature, they selected features using the Pearson correlation coefficient. Then, they clustered the logs with the K-means and DBSCAN algorithms. By measuring the salient cluster features, the clusters were associated with abnormal behaviors. Finally, they analyzed the abnormal clusters manually to determine the specific attack types.

### 4.4.3. Text Analysis-Based Detection

The text analysis-based detection regards logs as plain text. The methods utilize mature text processing techniques such as the n-gram to analyze logs. Compared with log feature extraction-based methods, this method understands log content at the semantic level and therefore has stronger interpretability.

In log-based detection, extracting text features from logs and then performing classification is the usual approach. When analyzing texts, a small number of keywords have large impacts on the whole text. Thus, the keywords in the field of cyber security aid in improving the detection effect. Uwagbole et al. [64] proposed an SQL-injection detection method for the Internet of Things (IoT). They collected and labeled logs from a real environment. The logs provide the contextual information of the SQL injection attack. First, they extracted 479,000 high-frequency words from the logs and then added 862 keywords that appear in SQL queries to compose a dictionary. Then, they removed duplicate and missing records from the log and balanced the data with SMOTE. Next, they extracted features using the n-gram algorithm and selected features using Chi-square tests. Finally, they trained an SVM model to perform classification, achieving accuracy, precision, recall, and F-measure scores of 98.6%, 97.4%, 99.7% and 98.5%, respectively.

In an actual network environment, normal samples are in the majority, and abnormal samples are rare. One-class classification, a type of unsupervised learning method, uses only normal samples for training, which solves the problem of a lack of abnormal samples. Vartouni et al. [65] proposed a web attack detection method based on the isolate forest model. They used the data of the CSIC 2010 dataset. First, they extracted 2572-dimensional features from HTTP logs with the n-gram. Then, they utilized an autoencoder to remove irrelevant features. Finally, they trained an isolation forest model to discover abnormal webs, which reached an accuracy of 88.32%.

### 5. Challenges and Future Directions

Table 5 lists papers on machine learning based IDSs which are introduced in this survey. It shows that deep learning methods have become a research hotspot (26 papers are listed, 14 papers adopt deep learning methods). KDD99 and NSL-KDD datasets are still widespread used. Although machine learning methods have made great strides in the field of intrusion detection, the following challenges still exist.

<div align="center">

**Table 5.** Methods and papers on machine learning based IDSs.

| Methods | Papers | Data Sources | Machine Learning Algorithms | Datasets |
|---|---|---|---|---|
| **Packet parsing** | Mayhew et al. [40] | Packet | SVM and K-means | Private dataset |
| | Hu et al. [41] | Packet | Fuzzy C-means | DARPA 2000 |
| **Payload analysis** | Min et al. [43] | Packet | CNN | ISCX 2012 |
| | Zeng et al. [44] | Packet | CNN, LSTM, and autoencoder | ISCX 2012 |
| | Yu et al. [45] | Packet | Autoencoder | CTU-UNB |
| | Rigak et al. [46] | Packet | GAN | Private dataset |
| **Statistic feature for flow** | Goeschel et al. [47] | Flow | SVM, decision tree, and Naïve Bayes | KDD99 |
| | Kuttranont et al. [48] | Flow | KNN | KDD99 |
| | Peng et al. [13] | Flow | K-means | KDD99 |
| **Deep learning for flow** | Potluri et al. [49] | Flow | CNN | NSL-KDD and UNSW-NB15 |
| | Zhang et al. [50] | Flow | Autoencoder and XGBoost | NSL-KDD |
| | Zhang et al. [51] | Flow | GAN | KDD99 |
| **Traffic grouping** | Teng et al. [52] | Flow | SVM | KDD99 |
| | Ma et al. [53] | Flow | DNN | KDD99 and NSL-KDD |
| **Statistic feature for session** | Ahmim et al. [54] | Session | Decision tree | CICIDS 2017 |
| | Alseiari et al. [55] | Session | K-means | Private dataset |
| **Sequence feature for session** | Yuan et al. [56] | Session | CNN and LSTM | ISCX 2012 |
| | Radford et al. [57] | Session | LSTM | ISCX IDS |
| | Wang et al. [58] | Session | CNN | DARPA 1998 and ISCX 2012 |
| **Rule-based** | Meng et al. [59] | Log | KNN | Private dataset |
| | McElwee et al. [60] | Log | DNN | Private dataset |
| **Log feature extraction with sliding window** | Tran et al. [61] | Log | CNN | NGIDS-DS and ADFA-LD |
| | Tuor et al. [62] | Log | DNN and RNN | CERT Insider Threat |
| | Bohara et al. [63] | Log | K-means and DBSCAN | VAST 2011 Mini Challenge 2 |
| **Text analysis** | Uwagbole et al. [64] | Log | SVM | Private dataset |
| | Vartouni et al. [65] | Log | Isolate forest | CSIC 2010 dataset |

</div>

**(1) Lack of available datasets**. The most widespread dataset is currently KDD99, which has many problems, and new datasets are required. However, constructing new datasets depends on expert knowledge, and the labor cost is high. In addition, the variability of the Internet environment intensifies the dataset shortage. New types of attacks are emerging, and some existing datasets are too old to reflect these new attacks. Ideally, datasets should include most of the common attacks and correspond to current network environments. Moreover, the available datasets should be representative, balanced and have less redundancy and less noise. Systematic datasets construction and incremental learning may be solutions to this problem.

**(2) Inferior detection accuracy in actual environments**. Machine learning methods have a certain ability to detect intrusions, but they often do not perform well on completely unfamiliar data. Most the existing studies were conducted using labeled datasets. Consequently, when the dataset does not cover all typical real-world samples, good performance in actual environments is not guaranteed—even if the models achieve high accuracy on test sets.

**(3) Low efficiency**. Most studies emphasize the detection results; therefore, they usually employ complicated models and extensive data preprocessing methods, leading to low efficiency. However, to reduce harm as much as possible, IDSs need to detect attacks in real time. Thus, a trade-off exists between effect and efficiency. Parallel computing [66,67] approaches using GPUs [48,68,69] are common solutions.

From summarizing the recent studies, we can conclude that the major trends of IDS research lie in the following aspects.

**(1) Utilizing domain knowledge**. Combining domain knowledge with machine learning can improve the detection effect, especially when the goal is to recognize specific types of attacks in specific application scenarios.

- The rule-based detection methods have low false alarm rates but high missed alarm rates include considerable expert knowledge. In contrast, the machine learning methods usually have high false alarm rates and low missed alarm rates. The advantages of both methods are complementary. Combining machine learning methods with rule-based systems, such as Snort [70–73], can result in IDSs with low false alarm rates and low missed alarm rates.
- For specific types of attacks, such as DOS [74–79], botnet [80], and phishing web [81], proper feature must be extracted according to the attack characteristics that can be abstracted using domain knowledge.
- For specific application scenarios, such as cloud computing [82,83], IoT [84–86], and smart grids [87,88], domain knowledge can be used to provide the environmental characteristics that are helpful in data collection and data preprocessing.

**(2) Improving machine learning algorithms**. Improvements in machine learning algorithms are the main means to enhance the detection effect. Thus, studies involving deep learning and unsupervised learning methods has an increasing trend.

- Compared with shallow models, deep learning methods learn features directly from raw data, and their fitting ability is stronger. Deep learning models with deep structures can be used for classification, feature extraction, feature reduction, data denoising, and data augmentation tasks. Thus, deep learning methods can improve IDSs from many aspects.
- Unsupervised learning methods require no labeled data; thus they can be used even when a dataset shortage exists. The usual approach involves dividing data using an unsupervised learning model, manually labeling the clusters, and then training a classification model with supervised learning [89–92].

**(3) Developing practical models**. Practical IDSs not only need to have high detection accuracy but also high runtime efficiency and interpretability.

- In attack detection, the real-time requirement is essential. Thus, one research direction is to improve the efficiency of machine learning models. Reducing the time required for data collection and storage is also of concern.
- Interpretability is important for practical IDSs. Many machine learning models, especially deep learning models, are black boxes. These models report only the detection results and have no interpretable basis [93]. However, every cyber security decision should be made cautiously. An output result with no identifiable reason is not convincing. Thus, an IDS with high accuracy, high efficiency and interpretability is more practical.

## 6. Conclusions

The paper first proposes an IDS taxonomy that takes data sources as the main thread to present the numerous machine learning algorithms used in this field. Based on this taxonomy, we then analyze and discuss IDSs applied to various data sources, i.e., logs, packets, flow, and sessions. IDSs aim to detect attacks, therefore it is vital to select proper data source according to attack characteristics. Logs contain detailed semantic information, which are suitable for detecting SQL injection, U2R, and R2L attacks. And packets provide communication contents, which are fit to detect U2L and R2L attacks. Flow represents the whole network environment, which can detect DOS and Probe attack. Sessions, which reflect communication between clients and servers, can be used to detect U2L, R2L, tunnel and Trojan attacks. For IDSs using these different data types, the paper emphasizes machine learning techniques (especially deep learning algorithms) and application scenarios.

Deep learning models are playing an increasingly important role and have become an outstanding direction of study. Deep learning approaches include multiple deep networks which can be used to improve the performance of IDSs. Compared with shallow machine learning models, deep learning models own stronger fitting and generalization abilities. In addition, deep learning approaches are independent of feature engineering and domain knowledge, which takes an outstanding advantage over shallow machine learning models. However, the running time of deep learning models are often too long to meet the real-time requirement of IDSs.

By summarizing the recent typical studies, this paper analyzes and refines the challenges and future trends in the field to provide references to other researchers conducting in-depth studies. Lacking of available datasets may be the biggest challenge. So unsupervised learning and incremental learning approaches have broad development prospects. For practical IDSs, interpretability is essential. Because interpretable models are convincing and can guide users to make a decision. The interpretability of models may become an important research direction about IDSs in the future.

**Author Contributions:** Writing—original draft preparation, H.L.; writing—review and editing, H.L.; writing—review and editing, B.L.

**Funding:** This research received no external funding

**Conflicts of Interest:** The authors declare no conflict of interest.

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
