# Peer review of "Machine Learning and Deep Learning Methods for Intrusion Detection Systems: A Survey"

_applsci, doi:10.3390/app9204396_

Round 1
Reviewer 1 Report
The paper surveys the research on machine learning techniques for intrusion detection. The paper can be useful for a beginner in the field.
There are many surveys (and books) on this topic and the paper fails in identifying a gap that it is trying to fill. The paper should explain why this new survey is necessary.
The first part of the paper (Sections 1-3) is very basic and can be found in many surveys on the topic.
Section 4 provides a good list of papers. I think that the paper should do some effort to better organize the list with respect to the classifications given in the previous sections.
Author Response
Thank you for the comments concerning our manuscript. All of these comments are very valuable and helpful to improve our paper, and have important guiding significances to our researches. We have studied the reviewers' comments carefully and have made revisions in responds to the comments, which we hope meet with your approval.
Q1: There are many surveys (and books) on this topic and the paper fails in identifying a gap that it is trying to fill. The paper should explain why this new survey is necessary.
Response: Thank you for the comments. We have added the gap in Section 1 (revised manuscript) to explain the meaning of our paper. We read many surveys on the topic, and found that almost all existing surveys classified research efforts by machine learning algorithms (in Section 1). These surveys emphasize machine learning algorithms rather than cyber security domain problems, which may cause confusions to cyber security researchers. For example, if a researcher wants to design an IDS, he/she should consider the attack types first rather than machine learning algorithms (Should I apply SVM or KNN?). In the paper, we propose a novel IDS classification method, taking the data source as the main classification consideration and treating the detection method as a secondary classification element (Figure 1). This taxonomy presents a path involving data–feature–attack behavior–detection model; when designing an IDS, a right logic is to define attack types first; then, find significant features of attacks; next, adopt proper data type according to features; finally, select proper detection methods. The taxonomy can help beginners to design an IDS step by step (from data collection to model building).
IDSs with Deep learning approaches update very fast, and this survey reflects the latest development in this field. The survey places emphasis on deep learning methods and provides a comparison with shallow machine learning models. Deep learning is an important branch of machine learning. At present, there are a few surveys on the topic which contains deep learning. But these surveys don’t include the latest research results, such as GAN. To our best knowledge, the paper is the first survey on this topic which introduces GRU and GAN. Compared with other surveys, this paper provides a more complete overview about the field. In addition, for this survey, we selected representative papers published from 2015 to 2019, which reflect the latest progress.
Moreover, this survey provides more comprehensive analysis and summary than previous surveys. The survey discusses challenges and future directions about this field. We mention interpretability problem of IDS. To our best knowledge, our paper is the first survey on the topic of model interpretability. Interpretability is important for IDSs. In cyber security, every decision should be made cautiously. The output result with identifiable reason is more convincing. Only the interpretable models can guide users to make a decision. Other challenges, such as lacking of available datasets, are also analyzed.
Q2: The first part of the paper (Sections 1-3) is very basic and can be found in many surveys on the topic.
Response: Thank you for the comments. The paper is intended for researchers who wish to begin research in the field of IDSs with machine learning methods. The first part of the paper (Sections 1-3) is basic but necessary for beginners.
In section 1, we introduce the meanings of IDSs and analyze the advantages of the IDSs with machine learning approaches.
In section 2, we introduce the concept of IDSs, and propose a new IDSs taxonomy basing on which we organize the introductions of the studies. We compare different IDSs and summarize their advantages and disadvantages (Table 1 and Table 2). When a researcher wants to develop an IDS, he/she can select a proper type of IDS to study according to Tables 1-2. The main function of surveys is to summarize and classify previous studies. We compare two existing IDS classification methods, detection-based method and data source-based methods. And we combine these two types of IDS classification methods, taking the data source as the main classification consideration and treating the detection method as a secondary classification element (Figure 1). We believe that the proposed taxonomy can make beginners know the current of the topic.
In section 3, we introduce common machine learning algorithms, metrics and benchmark datasets in IDS. We first introduce different machine learning algorithms and compare them (Table 3 and Table 4). Tables 3-4 can help researchers to choose suitable machine learning algorithms and provide improvement advises. And these algorithms are adopted in many researches (Section 4). So, these introductions can benefit beginners to read further. The metrics and benchmark datasets are basic knowledge of this topic.
Above all, the first part of the paper (Sections 1-3) provides the organization thread of the paper and important preliminary knowledge. The part of preliminary knowledge (Sections 2-3) is very basic, but these contents are necessary for researches on the topic. The first part is not the main body of the paper, but makes the readers easy to understand the follow-up parts of the paper.
Q3: Section 4 provides a good list of papers. I think that the paper should do some effort to better organize the list with respect to the classifications given in the previous sections.
Response: Thank you for the suggestion. We have added a table, i.e., Table 5 Methods and papers on machine learning based IDSs which contains list of papers introduced in Section 4. Table 5 is added in the beginning of Section 5, which summarizes the studies and discusses the challenges and future directions. All the paper appeared in Table 5 are representative and published in 2015 to 2019. The data source and machine learning methods of papers are listed in Table 5, which corresponds to the taxonomy we proposed (Figure 1).
Reviewer 2 Report
An honest survey, which has its merit. Specifically, I found the taxonomy depicted in Figure 1 to be quite interesting, as it provides a good categorization for the existing IDS techniques.
I would like to advise the authors to proofread the text - minor typos and sentence structure issues are found along the text (for instance, right at the beginning of the introduction: "Networks has increasing influences (...)".)
Also, the survey doesn't mention adversarial learning approaches (used for attacks against IDS, but also for improving the IDS detection accuracy) - quite recently, these approaches have been gaining visibility and I think authors should mention them in order to provide a more complete overview about the field.
Finally, I strongly advise for authors to improve the conclusions, which are one of the most disappointing aspects of this paper. A survey study should not be concluded in such a light tone, without providing a critical analysis of the techniques and tools which were discussed - albeit part of this critical analysis is already undertaken in the prior sections, the conclusion must synthesize the main points. Otherwise, I find the survey to be quite comprehensive - the kind of paper I would pass to a student of mine to introduce him/her to the field.
Author Response
Thank you for the comments concerning our manuscript. All of these comments are very valuable and helpful to improve our paper, and have important guiding significances to our researches. We have studied the reviewers' comments carefully and have made revisions in responds to the comments, which we hope meet with your approval.
Q1: I would like to advise the authors to proofread the text - minor typos and sentence structure issues are found along the text (for instance, right at the beginning of the introduction: "Networks has increasing influences (...)".)
Response: Thank you for the suggestion. We have proofread the text and the manuscript was edited by American Journal Experts (AJE).
Q2: Also, the survey doesn't mention adversarial learning approaches (used for attacks against IDS, but also for improving the IDS detection accuracy) - quite recently, these approaches have been gaining visibility and I think authors should mention them in order to provide a more complete overview about the field.
Response: Thank you for the suggestion. We have noticed that there are some researches which adopt adversarial learning approaches to improve IDS detection accuracy. We select the Generative Adversarial Network (GAN) as a representative of adversarial learning approaches. We introduce the relationship of GAN and adversarial learning in Section 3.1.2. GAN, a kind of novel deep learning approach, can be used for adversarial learning. GAN consists of a generator and a discriminator. The generator can produce the samples to confuse classifiers (adversarial samples). The adversarial learning is adding adversarial samples into training sets, which can improve the IDS detection accuracy. And we have adjusted sentences to emphasize the effect of adversarial learning.
In Section 4, we have mentioned 2 papers about adversarial learning. They are Rigaki et al. [46] and Zhang et al. [51].
Rigaki, M.; Garcia, S. Bringing a gan to a knife-fight: Adapting malware communication to avoid detection.2018 IEEE Security and Privacy Workshops (SPW). IEEE, 2018, pp. 70–75. Zhang, H.; Yu, X.; Ren, P.; Luo, C.; Min, G. Deep Adversarial Learning in Intrusion Detection: A Data Augmentation Enhanced Framework. arXiv preprint arXiv:1901.07949 2019.
In Rigaki et al. [46], they adopt GAN to generate malicious traffic that is similar to normal traffic. As the training epochs increased, the packets blocked by the IPS decreased and packet that passed inspection increased. Then, by analyzing the generated packets, the detection accuracy was improved. In Zhang et al. [51], they use GAN to generate new data which is similar to KDD99. Adding this generated data to the training set allows attack variants to be detected. They selected 8 types of attacks and compared the accuracies achieved on the original dataset compared to the expanded dataset. The experimental results showed that adversarial learning improved 7 accuracies in 8 attack types.
Q3: Finally, I strongly advise for authors to improve the conclusions, which are one of the most disappointing aspects of this paper. A survey study should not be concluded in such a light tone, without providing a critical analysis of the techniques and tools which were discussed - albeit part of this critical analysis is already undertaken in the prior sections, the conclusion must synthesize the main points. Otherwise, I find the survey to be quite comprehensive - the kind of paper I would pass to a student of mine to introduce him/her to the field.
Response: Thank you for the suggestion. We have added critical analysis and comparisons in the conclusions.
The paper proposes an IDS taxonomy that takes data sources as the main thread. So in the conclusion section, we first conclude and compare four data sources, i.e., logs, packets, flow, and sessions. We compare the main advantages and disadvantages of these cyber security data, and also point out for what kind of attack detection is certain type of data suitable for.
For machine learning algorithms, we focus on deep learning approaches. We summarize the benefits of deep learning, and also discuss its shortcomings compared with the shallow models.
In the last paragraph of the revised conclusion section, we synthesize the main points stated in the challenges and future direction section. We point out that, lacking of available datasets may be the biggest challenge, so unsupervised learning and incremental learning approaches have broad development prospects. For practical IDSs, interpretability is essential. Because interpretable models are convincing and can guide users to make a decision. The interpretability of models may become an important research direction about IDSs in the future.